# Potential UV-Protective Effect of Freestanding Biodegradable Nanosheet-Based Sunscreen Preparations in *XPA*-Deficient Mice

**DOI:** 10.3390/pharmaceutics14020431

**Published:** 2022-02-17

**Authors:** Tomomi Hatanaka, Khampeeraphan Ramphai, Shun Takimoto, Hiromi Kanda, Nami Motosugi, Minoru Kimura, Tomotaka Mabuchi, Midori Oyama, Tomoharu Takeuchi, Yosuke Okamura

**Affiliations:** 1School of Pharmacy, Faculty of Pharmacy and Pharmaceutical Sciences, Josai University, 1-1 Keyakidai, Sakado 350-0295, Japan; oyamami@josai.ac.jp (M.O.); t-take@josai.ac.jp (T.T.); 2Faculty of Medicine, Tokai University School of Medicine, 143 Shimokasuya, Isehara 259-1193, Japan; motosugi@is.icc.u-tokai.ac.jp (N.M.); mabuchi@is.icc.u-tokai.ac.jp (T.M.); 3Department of Applied Chemistry, School of Engineering, Tokai University, 4-1-1 Kitakaname, Hiratsuka 259-1292, Japan; khampeeraphan.rp@gmail.com; 4Course of Applied Science, Graduate School of Engineering, Tokai University, 4-1-1 Kitakaname, Hiratsuka 259-1292, Japan; takimoto.cosme@gmail.com (S.T.); for.univ.kanda@gmail.com (H.K.); 5The Institute of Medical Sciences, Tokai University, 143 Shimokasuya, Isehara 259-1193, Japan; kimura@is.icc.u-tokai.ac.jp

**Keywords:** xeroderma pigmentosum, UV protection, nanosheet, layered nanosheet, poly (L-lactic acid), avobenzone

## Abstract

Xeroderma pigmentosum (XP) is a rare autosomal recessive hereditary disorder. As patients with XP are deficient in nucleotide excision repair, they show severe photosensitivity symptoms. Although skin protection from ultraviolet (UV) radiation is essential to improve the life expectancy of such patients, the optimal protective effect is not achieved even with sunscreen application, owing to the low usability of the preparations. Nanosheets are two-dimensional nanostructures with a thickness in the nanometer range. The extremely large aspect ratios of the nanosheets result in high transparency, flexibility, and adhesiveness. Moreover, their high moisture permeability enables their application to any area of the skin for a long time. We fabricated preparations containing avobenzone (BMDBM) based on freestanding poly (L-lactic acid) (PLLA) nanosheets through a spin-coating process. Although monolayered PLLA nanosheets did not contain enough BMDBM to protect against UV radiation, the layered nanosheets, consisting of five discrete BMDBM nanosheets, showed high UV absorbance without lowering the adhesive strength against skin. Inflammatory reactions in *XPA*-deficient mice after UV radiation were completely suppressed by the application of BMDBM-layered nanosheets to the skin. Thus, the BMDBM layered nanosheet could serve as a potential sunscreen preparation to improve the quality of life of patients with XP.

## 1. Introduction

Xeroderma pigmentosum (XP) is a rare autosomal recessive hereditary disorder characterized by severe photosensitive and neurological symptoms [1,2,3]. Most patients with XP are deficient in nucleotide excision repair (NER) for DNA lesions induced by ultraviolet (UV) radiation [4]. This results in increased photosensitivity, leading to severe sunburn, freckling, xerosis, progressive development of pigmentation, and cancer in the sun-exposed area of the skin. Patients with XP often display skin cancer starting from early childhood, ocular diseases such as photophobia, and fetal neurodegeneration. The median age at death is 32 years, and 60% of premature deaths are caused by skin cancer [5]. XP is classified into eight subtypes by genetic mutation: NER-deficient types (XP-A to G) and variant type (XP-V), in which translesion DNA synthesis is impaired [6]. The frequency of XP-A is the highest in Japan, and the most severe clinical phenotypes are associated with dermal and neurological symptoms [7].

In the absence of a specific treatment targeting dysfunction in DNA repair systems, skin protection from UV radiation is essential to prevent the development of skin cancer and improve life expectancy. Sunscreens with a broad UV spectrum and high sun protection factor (SPF 50+ and PA+++), covering the whole body with clothes, gloves, and hats, wearing UV protective glasses and face shields, and attaching UV protective films to windows and lamps should be practiced to avoid sun exposure [8]. Despite the importance of UV protection, adherence varies among individuals. This can be explained by the different perceptions of the necessity of UV protection and its psychosocial impacts, such as changes in appearance and activity restrictions [9]. Unfortunately, even with sunscreen application, which is the most common form of UV protection, the optimal protective effect is not achieved because of several reasons: the applied amount is too small, some skin area is left without coverage, and reapplication is left too late [10,11,12]. Thus, there is an urgent need for the development of sunscreen preparations with excellent usability and sustainable protective effects that have negligible effects on appearance.

Various sunscreen preparations exist in liquid and semi-solid types, such as emulsions, lotions, gels, oily liquids, aerosols, and sticks. Unfortunately, they are weak in the presence of physical friction and can be easily removed when the skin is rubbed [13]. This necessitates reapplication and often causes uneven coating of sunscreen preparations. On the other hand, solid preparations such as dermal patches, tapes, plasters, and cataplasms, which are extensively used in topical and transdermal drug delivery [14,15], adhere to an area of the skin for long therapeutic periods. However, their application is limited to relatively flat skin surfaces, and they affect their appearance, owing to their complex configuration comprising a backing material, a drug-containing layer, and an adhesive layer. Thus, it is necessary to develop sunscreen preparations with multiple advantages: high frictional resistance, wide application area on skin, and negligible effect on appearance.

Ultrathin films, often called nanosheets, are two-dimensional nanostructures with a thickness in the nanometer order. Their extremely large aspect ratios result in high transparency, flexibility, and adhesiveness [16,17,18,19,20,21]. We previously reported that freestanding nanosheets composed of biodegradable and biocompatible polymers, such as poly (lactic acid) and poly (lactide-co-glycolide) acid, are promising vehicles for topical and transdermal drug delivery systems [22]. Their high adhesiveness and transparency are retained even when a drug is incorporated at the therapeutic level. Furthermore, their high moisture permeability enabled their application in any area of the skin for 12 h. If the polymeric nanosheets can incorporate sunscreens in the quantities required for UV protection, this will aid in the development of novel sunscreen preparations applicable to any part of the skin for a long time without impairing appearance. Ultimately, this will improve the quality of life of patients with XP.

The present study aimed to evaluate the potential UV-protective effects of freestanding biodegradable nanosheet-based sunscreen preparations in *XPA*-deficient mice. The preparations were fabricated through a spin-coating process using a water-soluble sacrificial membrane [18,20,21,22]. Poly (L-lactic acid) (PLLA) and poly (vinyl alcohol) (PVA) were used as the building blocks of the nanosheets and sacrificial layers, respectively. A UV absorber, avobenzone (4-*tert*-butyl-4′-methoxydibenzoylmethane, BMDBM) was included in the building blocks. When five-layered nanosheet preparations were developed, sodium alginate was added to the sacrificial layer as a gelling agent to obtain a preparation consisting of five discrete nanosheets [23]. The morphology, UV spectra, and stability of both types of nanosheet preparations were evaluated, and the photoprotective ability was assessed in *XPA*-deficient mice, an animal model of XP [24]. We hoped to develop a sunscreen preparation, which had advantage of nanosheets and UV protective ability of BMDBM, for XP patients.

## 2. Materials and Methods

### 2.1. Fabrication of BMDBM Nanosheet Preparations

#### 2.1.1. Materials

BMDBM and PLLA (molecular weight: 80,000–100,000, standard grade for medical devices) were purchased from Tokyo Chemical Industry Co. (Tokyo, Japan) and Polysciences Inc. (Warrington, PA, USA), respectively. PVA (molecular weight: approximately 22,000) and ethylenediaminetetraacetic acid (EDTA) were purchased from FUJIFILM Wako Pure Chemical Corporation (Osaka, Japan), and sodium alginate was purchased from Kanto Chemical Co. (Tokyo, Japan). Other chemicals and reagents were of analytical grade and used without further purification.

#### 2.1.2. Fabrication of Nanosheets and BMDBM Nanosheets

Nanosheets and those containing BMDBM were fabricated through a spin-coating-assisted layer-by-layer method using a PVA membrane as a sacrificial layer, which separated PLLA nanosheets from the silicon oxide (SiO_2_) substrate by dissolving it in water [18,20,21,22], as shown in Figure 1a. To develop a sacrificial layer, an aliquot of 1.0% PVA in ultra-purified water (0.5 mL) was dropped onto a SiO_2_ substrate (SIO wafer, 200 nm thick and made of SiO_2_, SUMCO Co., Tokyo, Japan), which was rotated at 4000 rpm for 20 s, using a spin-coater (MS-A100, Mikasa Co. Ltd., Tokyo, Japan), and then dried on a hotplate at 50 °C for 60 s. The sacrificial layer was spin-coated with 1.0% PLLA (0.5 mL) in dichloromethane at 4000 rpm for 20 s, followed by drying at 50 °C for 60 s, and a bilayered film was obtained on the SiO_2_ substrate. When nanosheet preparations containing BMDBM were fabricated, the UV absorber was dissolved in different concentrations of PLLA solution, ranging from 0.5–8.0%. The PLLA layer was spontaneously released from the substrate by immersion in water, scooped up with a nonwoven fabric composed of polyethylene and polypropylene (Daiso Industries Co., Hiroshima, Japan), and dried in a desiccator overnight. The resultant nanosheet preparations were covered with aluminum foil and stored in a desiccator at room temperature (approximately 20 °C) until use.

#### 2.1.3. Fabrication of Layered Nanosheets and BMDBM-Layered Nanosheets

Layered nanosheets consisting of five discrete nanosheets were fabricated as previously described, with a slight modification [23]. As mentioned above, a sacrificial layer was developed using an aqueous solution containing 1.8% sodium alginate and 0.25% PVA as a substitute for the 1% PVP aqueous solution. After drying at 50 °C for 90 s, the sacrificial layer was spin-coated with 0.7 mL of 1.0% PLLA in dichloromethane and dried again. In the fabrication of layered nanosheets containing BMDBM, a UV absorber was added to the PLLA spin-coating solution at a concentration of 0.5%. The layering process was repeated five times, and the final layer was coated with a sacrificial layer. The obtained multilayered substrate was immersed in a 2.0% calcium chloride aqueous solution at room temperature for 30 min to allow temporary gelation of the sacrificial layer and, consequently, the formation of a thick composite film. The composite film was isolated from the substrate by immersion in water. It was then scooped up using a nonwoven fabric and dried at 50 °C for 1 h. After the dried film was gently detached from the nonwoven fabric, only two sides of the film were fusion-cut sealed with POLYSEALER^®^ (PC-200, FUJIIMPALSE Co., Osaka, Japan) at 187 °C. The sealed film was then incubated in 0.5 M EDTA overnight, followed by incubation in distilled water at 37 °C for 1.5 h to dissolve and remove the sacrificial layers. The layered nanosheets and BMDBM-layered nanosheets were obtained by drying overnight in a desiccator at room temperature. After covering with aluminum foil, the layered nanosheet preparations were stored under the same conditions until use.

### 2.2. Morphology of BMDBM Nanosheet Preparations

#### 2.2.1. Macroscopic Morphology

The macroscopic morphology of the monolayered and five-layered nanosheets containing BMDBM was captured with a digital camera (M. ZUIKO DIGITAL 14–42 mm F3.5–5.6 II R, Olympus PEN Lite E-PL6, Olympus Co., Tokyo, Japan) immediately after floating on purified water.

#### 2.2.2. Microscopic Morphology

Cross-sectional images of the layered nanosheets were obtained using a field-emission scanning electron microscope (FE-SEM S-4800, Hitachi High-Technologies Co., Tokyo, Japan) at an accelerating voltage of 1 kV. A layered nanosheet was frozen in liquid nitrogen and then broken with tweezers. The broken pieces were fixed with carbon tape on the stage, and gold was sputtered with a desktop quick coater (SC-701, Sanyu Electron Co., Tokyo, Japan) at an ionic current of 3 mA for 30 s before observation.

#### 2.2.3. Measurement of Thickness

The thickness of the nanosheet preparations was measured using a stylus profilometer (DektakXT, Bruker Co., Berlin, Germany). Measurements were performed at least three times in different parts of the preparations.

### 2.3. Optical Analysis of the BMDBM Nanosheet Preparations

#### 2.3.1. Ultraviolet Visible Light Absorption (UV–Vis) Spectroscopy

UV-Vis spectra of the nanosheet preparations were acquired using a UV-Vis spectrophotometer UV-2600 (Shimadzu, Kyoto, Japan). After water spraying, the preparation was adhered to a quartz plate of 2.0 mm thickness and dried overnight in a desiccator prior to analysis.

#### 2.3.2. Fourier Transform Infrared (FT-IR) Spectroscopy

Residual sodium alginate and PVA in the layered nanosheet preparations were detected using an FT-IR spectrometer (FTIR-8400, Shimadzu). Before the FT-IR analysis, a nanosheet preparation was applied to a substrate composed of calcium fluoride, which had a diameter of 2.0 mm and a thickness of 2 mm, by spraying water. Then, it was dried overnight in a desiccator prior to analysis.

### 2.4. Photoprotective Ability of the BMDBM Nanosheet Preparations

#### 2.4.1. Mice

Homozygous *XPA*-deficient mice [24] with a Hos:HR-1 hairless mouse background were obtained from breeding stocks maintained at the Center of Genetic Engineering for Human Diseases at Tokai University School of Medicine. The stock was derived from embryos stored at the Central Institute for Experimental Animals (Kawasaki, Japan). Five male mice, aged 8–9 weeks, were used to assess the photoprotective effect of BMDBM-layered nanosheets. In the other five mice, layered nanosheets were applied without BMDBM.

#### 2.4.2. UV Irradiation

*XPA*-deficient mice were anesthetized with isoflurane. After water spraying, the BMDBM-layered nanosheets of 4 × 4 cm^2^ and 3 × 3 cm^2^ were attached to the right dorsal skin and the entire right ear, respectively. Layered nanosheets without BMDBM were applied to the other mice in the same manner. A bank of four UVB lamps (G15T8E, Sankyo Denki Co., Hiratsuka, Japan), which emit UV rays between 280–360 nm, with a peak emission at 306 nm, was used to irradiate the mice. The mice were placed 25 cm below the light source in an abdominal position and were irradiated only once using UVB (0.5 kJ/m^2^), which was set using a UV ray intensity meter (UV-340C, CUSTOM Co., Tokyo, Japan).

#### 2.4.3. Observation of Inflammatory Reactions

Inflammatory reactions in *XPA*-deficient mice were assessed just before UV irradiation and 24, 48, 72, and 96 h later. Under isoflurane anesthesia, a visual inspection of the entire body skin was conducted, and the features were captured using a digital camera. Ear thickness was measured using a digital micrometer (19995, Shinwa Rules Co., Tsubame, Japan) as an index of intracellular edema.

#### 2.4.4. Histological Analysis

For histological analysis, mice were sacrificed by cervical dislocation immediately after the final visual inspection and ear thickness measurements. Then, the skin was freshly excised, fixed with 10% neutralized formalin overnight at room temperature, and embedded in paraffin. The paraffin-embedded skin was sliced into 4 µm thick vertical sections, deparaffinized in xylene, rehydrated in an ethanol gradient, and stained with hematoxylin-eosin. The examination was carried out using a stereomicroscope (SMZ25; Nikon Co., Tokyo, Japan).

#### 2.4.5. Statistics

Differences between two groups were assessed for significance using the Student’s *t*-test. Statistical significance was set at *p* < 0.05.

## 3. Results

### 3.1. BMDBM Nanosheet Preparations

#### 3.1.1. BMDBM Nanosheets

PLLA nanosheet preparations containing BMDBM were fabricated through a spin-coating-assisted layer-by-layer method using a PVA membrane as the sacrificial layer (Figure 1a). The main characteristics of the nanosheet preparations are listed in Table 1. Although the thickness of the BMDBM nanosheets was 77.0 ± 1.8 nm, which was slightly more than that of the nanosheets without the UV absorber, the high transparency, flexibility, and adhesiveness of the nanosheets were maintained (Figure 2a). A sufficiently low absorbance was observed in the visible wavelength range (400–600 nm), reflecting the high transparency of the sheets (Figure 2e). The absorbance peak at 363 nm in the spectra was assigned to BMDBM, but the absorbance in the UV wavelength range was unfortunately insufficient for UV protection. The concentration of BMDBM added to the spin-coating solution was increased to 8%. However, the higher BMDBM concentration resulted in an increase in the thickness of the preparations (Figure 2d), which reduced their adhesiveness.

#### 3.1.2. BMDBM-Layered Nanosheets

BMDBM-layered nanosheets consisting of five discrete nanosheets were developed (Figure 1b). Sodium alginate was added to the sacrificial layers as a gelling agent, and cross-linking of the gels by calcium ions prevented the adhesion of the nanosheets. After the two sides of all the layers were fusion-cut sealed together, the sacrificial layers were dissolved in water and removed using EDTA. A slight peak near 3500 cm^−1^, derived from sodium alginate, was observed in the Fourier transform infrared spectra of the layered nanosheets (Figure 2f). Although the BMDBM-layered nanosheet could be handled as a single sheet (Figure 2b), an interior space existed between the five layers (Figure 2c). The thickness was 451.3 ± 71.4 nm, which was five times more than the thickness of the monolayered nanosheet (Table 1). This indicates that high adhesiveness was maintained in the monolayered nanosheets. UV absorbance was increased by developing a layered nanosheet, which hardly changed upon being shaded until 30 days after fabrication (Figure 2e).

### 3.2. UV Protective Effect in XPA-Deficient Mice

The effect of BMDBM-layered nanosheets on inflammatory reactions after UV exposure was evaluated in *XPA*-deficient mice. After the preparations were applied to the right dorsal skin and the whole right ear, mice were irradiated with UVB at 0.5 kJ/m^2^ once. Layered nanosheets without BMDBM were applied to the other mice in a similar manner. The nanosheet preparations were almost invisible immediately after application and were barely recognized owing to the light reflection at an observation angle; then, they completely disappeared the next day (Figure 3). In the application of nanosheets without BMDBM, erythema was observed in the skin area with and without layered nanosheet application 24 h after UV exposure. Dermatitis was most severe at 48 h but began to resolve at 72 h, when ear swelling and dorsal skin exfoliation were observed. Such inflammatory reactions were not observed in the skin area where the BMDBM-layered nanosheet was applied.

The ear thickness of mice increased over time, even if the ear was covered with a layered nanosheet without BMDBM (Figure 4a). In contrast, application of BMDBM-layered nanosheets maintained the ear thickness, whereas the uncovered ears became thicker, similar to the layered nanosheet without BMDBM. Histological examination of the dorsal skin was performed using hematoxylin and eosin after 96 h of UV exposure (Figure 4b). Exfoliation and dense infiltration of inflammatory cells, such as neutrophils, lymphocytes, and plasma cells, were detected in both the untreated skin and the skin area where the layered nanosheet was applied. However, the skin where the BMDBM-layered nanosheet was applied showed a histological structure similar to that of the skin of mice not subjected to UV irradiation (Figure 4c).

## 4. Discussion

Research on the biomedical application of nanosheets has increased over the past ten years, owing to the rapid progress in fabrication techniques [25]. The use of biodegradable polymers, together with the isolation of nanosheets from substrates, has expanded their application in novel drug delivery systems, wound treatment bandages, and biosensing devices. The high adhesiveness, transparency, and moisture permeability of freestanding biodegradable nanosheets are beneficial characteristics for both topical and transdermal drug delivery systems [22]. Although the medical application of biodegradable polymers has been studied since the 1960s, commercial pharmaceutical preparations represent only an implanted sustained-release preparation of the luteinizing hormone-releasing hormone. The use of biodegradable polymers for pharmaceutical preparations has attracted attention because polymers made from renewable resources, such as agriculture or marine products, result in less greenhouse gas emissions [26].

In the present study, freestanding biodegradable nanosheets were applied to sunscreen preparations for the photoprotection of *XPA*-deficient mice (an established experimental model of XP in humans). PLLA was used as the building block for the nanosheets. Poly (lactic acid), including PLLA, poly (glycolic acid), and their copolymers are clinically used as bone fixation devices, such as plates, screws, and pins, and as scaffolds for soft and hard tissue repair. These polymers showed satisfactory biocompatibility and the absence of significant toxicity [27]. PLLA degrades slowly in vivo by hydrolysis into lactic acid oligomers and monomers, and the biodegradation rate changes depending on its molecular weight and specific surface area, environmental temperature, and pH, as well as the presence of enzymes such as proteinase K [28,29]. We previously reported that PLLA nanosheets were sufficiently stable for at least three weeks under simulated gastric conditions (0.5% pepsin at pH 1.0) [18]. It appears that the PLLA nanosheets are stable enough for daily application to the skin.

Whether ultra-thin nanosheets can carry adequate levels of UV absorbers is the most important concern in the development of sunscreen preparations. In preliminary studies, nanosheets were prepared using several UV absorbers, such as oxybenzone-3 (2-hydroxy-4-methoxybenzophenone), oxybenzone-4 (2-hydroxy-4-methoxybenzophenone-5-sulfonic acid), and BMDBM. When the UV absorbers were added to the spin-coating solution at half of the PLLA concentration (Figure 1a), the UV absorbance of the preparations with BMDBM was the highest among the tested UV absorbers (Appendix A). The molar absorption coefficient of BMDBM in the UVB wavelength region was higher than that of benzophenones. Some hydrophilic UV absorbers were insoluble in dichloromethane, which is the solvent of the spin-coating solution of PLLA. The concentration of oxybenzone-4 in the solution could not be higher than 0.5%, as high oxybenzone-3 concentrations prevented the film formation of nanosheets. In the spin-coating assisted layer-by-layer self-assembly method, nanosheets are formed through centrifugal force and air shear force [30]. In the formation process, the adsorption and rearrangement of helical PLLA chains and UV absorbers on the surface of the PVA sacrificial membrane are almost simultaneously achieved at a high spinning speed for a short time. The affinity for PLLA and the molar absorption coefficient of the UV absorber are critical factors in determining the UV protective ability of nanosheet preparations.

Unfortunately, the UV absorbance of the BMDBM nanosheet was significantly lower than that of the commercial sunscreen preparations (Figure 2e and Appendix A). This problem was solved by adding BMDBM to the spin-coating solution at a high concentration. In addition to increasing the UV absorbance of the nanosheet preparations, the thickness of the preparation also increased (Figure 2d). It was known that the thickness of nanosheets was controlled by concentration of solutes in the spin-coating solutions [18]. It was reported that a PLLA nanosheet less than 100 nm thick has a lower glass transition temperature and elastic modulus than those of the bulk PLLA, owing to decreased interactions between polymer chains in the thin film [18,31]. Softness and flexibility increase the ability to adhere to various types of surfaces, such as glass, steel, plastic, and organ, including the skin. However, the adhesive strengths were remarkably decreased for nanosheets thicker than 100 nm. Thus, the thickness of the preparation should be less than 100 nm to maintain high adhesion to the skin surface.

Accordingly, BMDBM-layered nanosheets consisting of 3, 5, and 10 discrete nanosheets were fabricated (Figure 1b). Although the UV absorbance of the three-layered preparation was not sufficient for UV protection, that of the five-layered preparation was higher than that of the commercial sunscreen preparations and remained unchanged for 30 days under shaded conditions (Figure 2e and Appendix A). As the edges of the preparation were efficiently fusion-cut sealed at 182 °C, which is higher than the melting temperature of PLLA (166 °C), the preparation could be handled as a single sheet (Figure 2b). Despite the increased total thickness (Table 1), the preparations stably adhered to the skin and were almost invisible (Figure 3). When a preparation is applied on the skin surface after water spraying, the water penetrates between the individual nanosheet layers, and thus each layer is separated by dissolving residual alginate and PVA [23]. The first layer was directly adhered to the skin by water evaporation, and the subsequent layers adhered to it one after another. Notably, the preparation appeared almost as invisible. Changes in appearance remarkably affect the adherence of patients with XP to UV-protective preparations [9]. The ten-layered preparation was visualized based on the irregular reflection of light owing to uneven lapping of the nanosheets. The nanosheet preparations were removed by grooming mice in the present study. In our previous study, however, a nanosheet preparation was left on the skin of human subjects for 12 h until washing with soap and water [22]. The five-layered PLLA nanosheets were more difficult to burst than monolayered nanosheets [23]. The BMDBM-layered nanosheets were sunscreen preparations with high adhesive strength and high UV absorbance, without any effect on appearance.

The UV protective effects of BMDBM-layered nanosheets were assessed in *XPA*-deficient mice [24]. These mice display no obvious physical abnormalities or pathological aberrations, but are defective in NER and are highly photosensitive; hence, they are extensively used as an in vivo model of XP [32,33]. Erythema, ear swelling, exfoliation, and dense infiltration of inflammatory cells were observed in *XPA*-deficient mice after irradiation with UVB at 0.5 kJ/m^2^ (Figure 3 and Figure 4), which is lower than that required to induce inflammatory reactions in other types of mice [34,35]. Such inflammatory reactions were completely suppressed by the application of the BMDBM-layered nanosheets. The skin where the BMDBM-layered nanosheet was applied showed a histological structure similar to that of the skin of mice not subjected to UV irradiation. *XPA*-deficient mice may show inflammation during breeding. It can be concluded that the BMDBM-layered nanosheet might represent a potential sunscreen preparation to improve the quality of life of patients with XP.

Although only BMDBM was included in the present preparations, the combined use of a few UV absorbers is desirable to shield a wide range of UV radiation wavelengths. Depending on the composition of the preparations, the UV absorbance may have to be adjusted by adjusting the number of nanosheet layers in the preparation. The relationship between the UV protective effect of the preparation and the UV irradiation dose must be clarified, and the SPF and PA of the preparation should be tested for practical use [36]. In particular, the method of application to the skin can be greatly improved. The application of the preparation to the skin after water spraying is complicated and unsuitable for daily use. In future studies, we plan to develop improved application methods with the aid of spray preparation. If an excellent preparation can be developed, it may be employed in the cosmetic industry. The BMDBM-layered nanosheet reported in the present study lays the foundation for the development of new types of sunscreen preparations.

## 5. Conclusions

The BMDBM-layered nanosheet preparation represents a promising strategy for UV protection in patients with XP because of its high UV protective ability, frictional resistance, wide application area on skin, and negligible effect on appearance. The preparation method can be improved by the development of spray preparations. The preparation can contain other sunburn therapeutic drugs, including steroids and antimicrobials, if necessary, as well as UV absorbers. The improved preparations may enhance the quality of life of not only patients with XP but also healthy individuals.

## Figures and Tables

**Figure 1 pharmaceutics-14-00431-f001:**
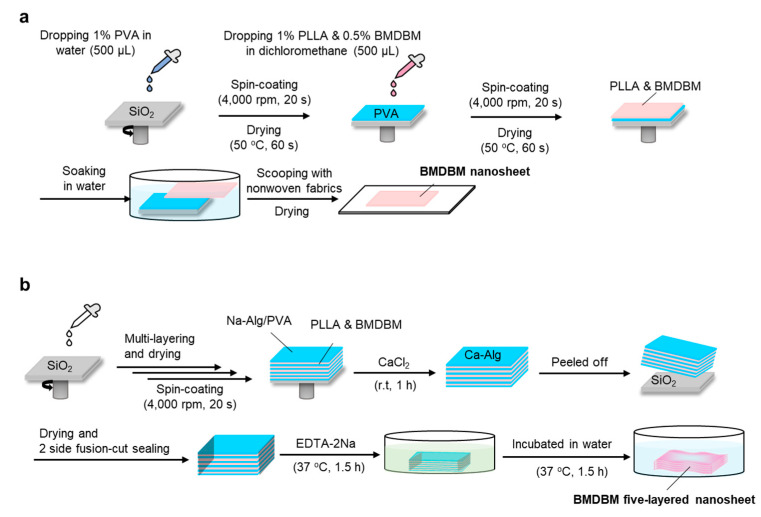
Schematic for preparation of BMDBM nanosheets. (**a**) BMDBM nanosheets fabricated via a spin-coating method using a sacrificial membrane. (**b**) BMDBM five-layered nanosheet fabricated utilizing sodium alginate as a gelling agent for the sacrificial membrane. PLLA, poly (L-lactic acid); BMDBM, 4-*tert*-butyl-4′-methoxydibenzoylmethane; PVA, poly (vinyl alcohol); Na-Alg, sodium alginate; Ca-Alg, calcium alginate, EDTA-2Na, ethylenediaminetetraacetic acid disodium salt.

**Figure 2 pharmaceutics-14-00431-f002:**
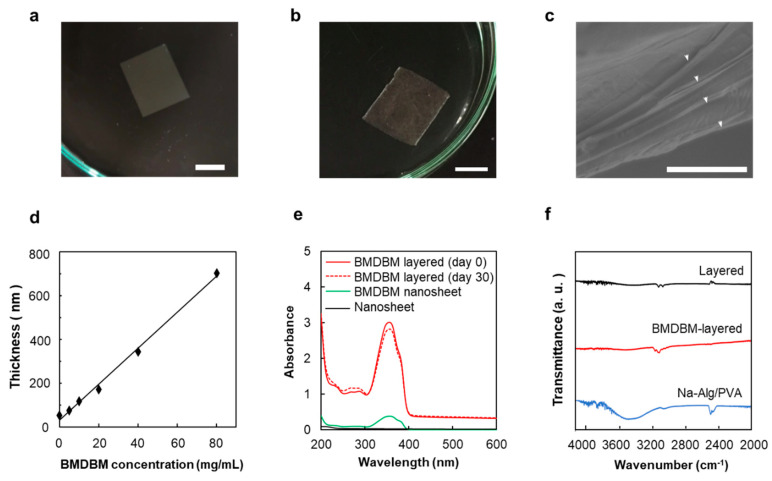
Characterization of BMDBM nanosheet preparations. (**a**) Macroscopic image of a BMDBM nanosheet floating in water. Scale bar represents 1.0 cm. (**b**) Macroscopic image of a BMDBM-layered nanosheet floating in water. Scale bar represents 1.0 cm. (**c**) Sectional scanning electron microscope image of a layered nanosheet. Arrow heads indicate the discrete nonosheet layers. Scale bar represents 3.0 µm. (**d**) Relationship between the thickness of nanosheet and BMDBM concentration. (**e**) Ultraviolet visible light absorption spectra of nanosheet preparations. (**f**) Fourier transform infrared spectra of nanosheet preparations. Each value represents the mean ± standard deviation of 3–5 measurements. BMDBM, 4-*tert*-butyl-4′-methoxydibenzoylmethane (avobenzone); Na-Alg, sodium alginate; PVA, poly (vinyl alcohol).

**Figure 3 pharmaceutics-14-00431-f003:**
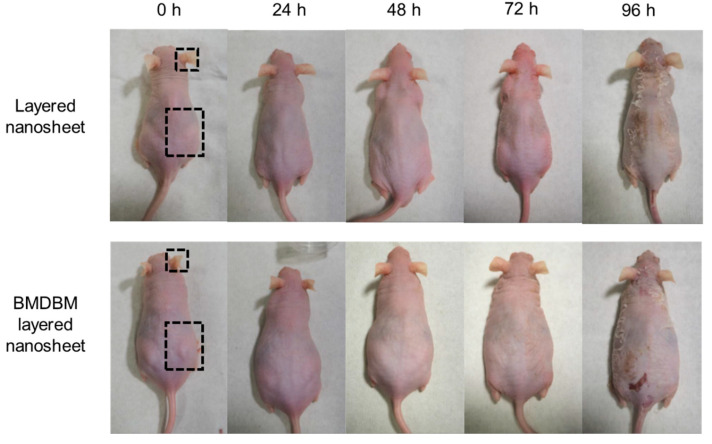
Protective effect of BMDBM-layered nanosheets on the appearance of *XPA*-deficient mice after UV exposure. After application of BMDBM-layered nanosheets or layered nanosheets (without BMDBM) on the area surrounded with dotted lines in the ear and the dorsal skin, mice were irradiated with UVB at 0.5 kJ/m^2^ only once. The results were representative examples of five mice in each group.

**Figure 4 pharmaceutics-14-00431-f004:**
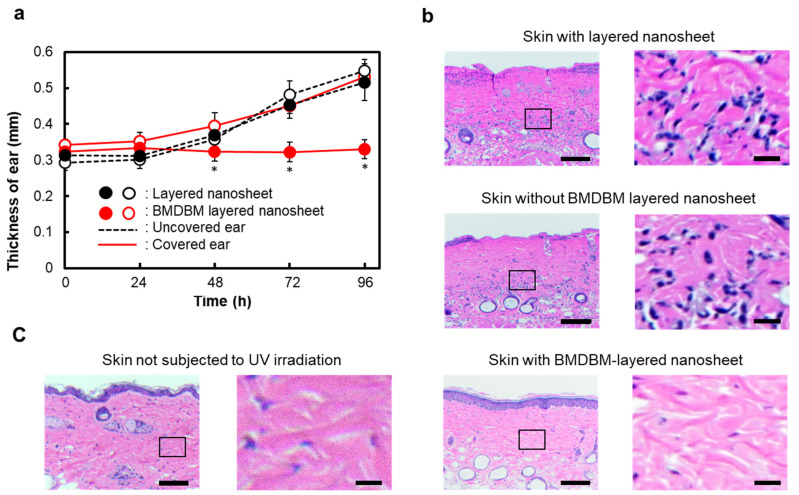
Protective effect of the BMDBM-layered nanosheet against ear swelling and pathological findings in *XPA*-deficient mice after UV exposure. After application of the BMDBM-layered nanosheet or layered nanosheet on the right ear and a part of the right dorsal skin, mice were irradiated with UVB at 0.5 kJ/m^2^ once. (**a**) Ear swelling after UV exposure. Each point represents the mean ± standard deviation of data obtained from five mice. * *p* < 0.05 when covered right ear with BMDBM-layered nanosheet was compared to uncovered left ear. (**b**) Histological analysis of dorsal skin using hematoxylin and eosin 96 h after UV exposure. Images in the right column show an enlarged view of the marked area in the corresponding images in the left column. Scale bars represent 500 and 125 µm for the left and right columns, respectively. Upper and middle images show exfoliation of the skin and filtration of inflammatory cells stained with hematoxylin eosin blue. (**c**) Histological analysis of dorsal skin not subjected to UV irradiation using hematoxylin and eosin.

**Table 1 pharmaceutics-14-00431-t001:** Characteristics of BMDBM nanosheet preparations.

Preparation	Building Blocks		Sacrificial Layers		Final Thickness ^e^ (nm)
	PLLA ^a^ (% *w*/*v*)	BMDBM ^b^ (% *w*/*v*)	PVA ^c^ (% *w*/*v*)	Na-Alg ^d^ (% *w*/*v*)	
Nanosheet	1.0	−	1.0	−	58.7 ± 4.1
BMDBM nanosheet	1.0	0.5	1.0	−	77.0 ± 1.8
Layered nanosheet	1.0	−	1.8	2.5	360.0 ± 4.0
BMDBM-layered nanosheet	1.0	0.5	1.8	2.5	451.3 ± 71.4

^a,b,c,d^: Concentrations of poly (L-lactic acid), 4-*tert*-butyl-4′-methoxydibenzoylmethane (avobenzone), poly (vinyl alcohol), and sodium alginate in the spin-coating solutions, respectively. ^e^: Each value represents the mean ± standard deviation of 3–5 measurements.

## Data Availability

Not applicable.

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
