# Peer review of "Potential UV-Protective Effect of Freestanding Biodegradable Nanosheet-Based Sunscreen Preparations in XPA-Deficient Mice"

_pharmaceutics, 2022, doi:10.3390/pharmaceutics14020431_

Round 1
Reviewer 1 Report
Hatanaka et al. have demonstrated the use of a avobenzene (BMBDM) based layered nanosheet to be used as a potential sunscreen for patients suffering from Xerorderma pigmentosum. Overall, the idea is super-interesting and the authors have demonstrated good phyisicochemical and functional characterization of the constructed scaffold. However, a few comments must be addressed before the manuscript is deemed suitable for publication,
- Is there a specific rationale for going with five layered structure? Is it for the mechanical strength or is it for enough UVB protectiveness? Authors should clarify the rationale in the discussion section.
- Authors have claimed biodegradability of the scaffold but have provided no experimental data supporting the same. While the polymers themselves are biodegradable, authors should provide evidence of the scaffold disintegrating after 'x' time under aqueous conditions.
- In section 3.2, authors mention the "control mice group". Does this refer to the regions on the same mice not exposed to the scaffolds or was this group different from the ones shown in Figure 3? If this was a different group, authors should that in Figure 3.
- While the H&E images in Figure 4, indicate some level of inflammation, it is itself not conclusive evidence. Authors should consider performing immunohistochemistry analysis to look for inflammatory biomarkers, which would allow them to substantiate their claims for erythmea.
Author Response
Thank you very much for your helpful comments on our manuscript. Our replies to your comments are shown below.
- Is there a specific rationale for going with five layered structure? Is it for the mechanical strength or is it for enough UVB protectiveness? Authors should clarify the rationale in the discussion section.
Response: We fabricated BMDBM-layered nanosheets consisting of 3, 5, and 10 discrete nanosheets. When choosing the final preparation, we primarily considered the UV protection ability, although the appearance and mechanical strength were also considered. This information has been added to L. 423-447.
- Authors have claimed biodegradability of the scaffold but have provided no experimental data supporting the same. While the polymers themselves are biodegradable, authors should provide evidence of the scaffold disintegrating after 'x' time under aqueous conditions.
Response: PLLA nanosheets were stable for at least three weeks under aqueous conditions. Information on the biodegradation of PLLA nanosheets has been added to L. 371-385.
- In section 3.2, authors mention the "control mice group". Does this refer to the regions on the same mice not exposed to the scaffolds or was this group different from the ones shown in Figure 3? If this was a different group, authors should that in Figure 3.
Response: We intended to refer to the areas on the same mice where the preparation was not applied. The term “control mice” is confusing. Therefore, Sections 2.4 and 3.2 have been revised to omit this term (L. 222-224, L. 229-231, L. 314-315, L. 318-320, and L. 332-336).
- While the H&E images in Figure 4, indicate some level of inflammation, it is itself not conclusive evidence. Authors should consider performing immunohistochemistry analysis to look for inflammatory biomarkers, which would allow them to substantiate their claims for erythmea.
Response: As you mentioned, immunohistochemical analysis using inflammatory biomarkers should be performed. XPA-deficient mice may show inflammation during breeding. Therefore, HE images of the mouse skin that was not subjected to UV irradiation are shown in Figure 4 (L. 340-343 and L. 457-460). The HE images support the UV protective effects, together with ear swelling and the macroscopic images. We want to perform the immunohistochemical analysis in our next report.
Reviewer 2 Report
The authors achieved a highly interesting manuscript entitled [Potential UV-protective Effect of Freestanding Biodegradable Nanosheet-based Sunscreen Preparations in XPA-deficient Mice]. The authors concluded that [Sunscreen preparations based on freestanding biodegradable nanosheets represent a promising strategy for UV protection in XP patients. Although monolayered PLLA nanosheets did not contain enough BMDBM to protect against UV radiation, the layered nanosheets consisting of five discrete BMDBM nanosheets showed high UV absorbance without lowering the adhesive strength to the skin surface. Moreover, the BMDBM-layered nanosheets did not affect the appearance of XPA-deficient mice owing to their high transparency. Further studies are needed to improve the UV protective ability and the usability of the preparation. The improved preparations may enhance the quality of life of not only XP patients but also the healthy individuals]……….
I have a major concern for the in vivo model, where the authors did not provide images of control mice subjected to UV only and that were not subjected to UV.
Author Response
Thank you very much for bringing these important points to our attention. Our responses to your comments are shown below.
I have a major concern for the in vivo model, where the authors did not provide images of control mice subjected to UV only and that were not subjected to UV.
Response: The descriptions in Sections 2.4 and 3.2 may have led to confusion; therefore, these sections were rewritten (L. 222-224, L. 229-231, L. 314-315, L. 318-320, and L. 332-336). As the preparations were not applied to the left side of the body, this side was directly subjected to UV irradiation. According to your comment, HE images of the mouse skin that had not been subjected to UV irradiation were added to Figure 4 (L. 340-343 and L. 457-460).
Reviewer 3 Report
The manuscript entitled 'Potential UV-protective Effect of Freestanding
Biodegradable Nanosheet-based Sunscreen Preparations
in XPA-deficient Mice ' is presented very well. The research plan was very good and results were discussed very clearly.
Author Response
Thank you very much for your review and favorable comments on our manuscript.
Reviewer 4 Report
Title: Potential UV-protective Effect of Freestanding Biodegradable Nanosheet-based Sunscreen Preparations in XPA-deficient Mice
Recommendation: Major revision
Comments:
In this manuscript, Tomomi Hatanaka and coworkers were studied on the five discrete BMDBM nanosheets to treat xeroderma pigmentosum. It was found that the five discrete BMDBM nanosheets showed high UV absorbance to the skin surface. Moreover, the five discrete BMDBM nanosheets enhanced the quality of life of not only XP patients but also the healthy individuals. This work is interesting, but some details need to be improved and supplemented. There are some issues need to be addressed:
- This study is focused on treating XP patients. What is the specific mechanisms of BMDBM-layered nanosheets should be described in introduction.
- Accordingto Figure 2d, we can include that higher BMDBM concentration resulted in an increase in the thickness, please describe the detailed mechanism of this phenomenon.
- In Figure 2f, there is not obvious difference between the curve of Layered and BMDBM layered. Thus, it is recommend that the mark the signal of special peaks.
- In Figure 3, the quantitative area data were recommend provided to compare the protective effect.Additionally, it is inaccurate results in Figure 3, because the amount of mouse in each group is below three.
- In Figure 4b, thedetailed images of histological analysis cannot be observed clearly, leading to the protective effect of prepared nanosheets is doubtful.
- According to the introduction, the adhesiveness of nanosheets is a crucialfactor, so it is necessary to demonstrate the viscosity of nanosheets.
- The manuscript would benefit from careful read through.There are a number of grammatical errors in the manuscript such as para 2 of introduction “left without coverage”, Table 1 “BMDBM nanosheet preparations” should be changed as “BMDBM nanosheets”. The authors should strictly revise the language of this manuscript.
Author Response
Thank you very much for your helpful comments on our manuscript. Our replies to your comments are shown below.
- This study is focused on treating XP patients. What is the specific mechanisms of BMDBM-layered nanosheets should be described in introduction.
Response: We hoped to develop a sunscreen preparation, which has advantage of nanosheets and UV protective ability of BMDBM, for XP patients. Our corresponding statement was added to L. 113-115.
- According to Figure 2d, we can include that higher BMDBM concentration resulted in an increase in the thickness, please describe the detailed mechanism of this phenomenon.
Response: Relationship between thickness of nanosheets and concentration of solutes in spin-coating solutions has been investigated. Corresponding comment was added to L. 413-415, together with referring to a report.
- In Figure 2f, there is not obvious difference between the curve of Layered and BMDBM layered. Thus, it is recommend that the mark the signal of special peaks.
Response: Because FT-IR spectra in Figure 2f showed the presence of sodium alginate but not BMDBM, they were not obviously different between layered and BMDBM layered nanosheets.
- In Figure 3, the quantitative area data were recommend provided to compare the protective effect. Additionally, it is inaccurate results in Figure 3, because the amount of mouse in each group is below three.
Response: It was difficult to obtain the quantitative area data, because the mice were partially applied nanosheet preparations (L. 227-231). The results in Figure 3 were representative examples of five mice in each group (L. 222-224). The statement was added to legends of Figure 3.
- In Figure 4b, the detailed images of histological analysis cannot be observed clearly, leading to the protective effect of prepared nanosheets is doubtful.
Response: XPA-deficient mice may show inflammation during breeding. Accordingly, HE images of the mouse skin that had not been subjected to UV irradiation were added to Figure 4 (L. 340-343 and L. 457-460). The HE images support the UV protective effects, together with ear swelling and the macroscopic images.
- According to the introduction, the adhesiveness of nanosheets is a crucial factor, so it is necessary to demonstrate the viscosity of nanosheets.
Response: As you mentioned, the adhesiveness of nanosheets is a crucial factor. The mechanisms of adhesion have been studied, so that corresponding comments were added to L. 441-445, together with referring to previous reports.
- The manuscript would benefit from careful read through. There are a number of grammatical errors in the manuscript such as para 2 of introduction “left without coverage”, Table 1 “BMDBM nanosheet preparations” should be changed as “BMDBM nanosheets”. The authors should strictly revise the language of this manuscript.
Response: Our manuscript has already undergone the English proofreading by the native speakers (L. 507-508).
Reviewer 5 Report
This report (pharmaceutics-1579577) entitled “Potential UV-protective Effect of Freestanding Biodegradable Nanosheet-based Sunscreen Preparations in XPA-deficient Mice” by Tomomi Hatanaka seems interesting to the journal readers. This report is on protective agents against xeroderma pigmentosum (XP) which is a rare autosomal recessive hereditary disorder. This seems novel and interesting method and materials is reported. I recommend this publication after minor revision.
Specific comments:
* Materials and Methods; suggestion to check the toxicity of Nanosheets, and Layered Nanosheets.
*Results: Discuss the result of toxicity in vitro and in vivo.
* Conclusion; do not repeat the details of an abstract, mention practical relevance, future implications.
Author Response
Thank you very much for your helpful comments on our manuscript. Our replies to your comments are as shown below.
* Materials and Methods; suggestion to check the toxicity of Nanosheets, and Layered Nanosheets.
*Results: Discuss the result of toxicity in vitro and in vivo.
Response: We used standard-grade PLLA for medical devices (L. 119-120). Information on the toxicity of the PLLA nanosheets has been added to L. 371-385.
* Conclusion; do not repeat the details of an abstract, mention practical relevance, future implications.
Response: According to your comment, section 5 has been thoroughly revised (L. 479-486).
Round 2
Reviewer 2 Report
The authors clarified my comments.
Reviewer 4 Report
Title: Potential UV-protective Effect of Freestanding Biodegradable Nanosheet-based Sunscreen Preparations in XPA-deficient Mice
Pharmaceutics
Recommendation: Accept
Comments:
In this revised manuscript, Tomomi Hatanaka and coworkers have completely described the specific mechanism of the five discrete BMDBM nanosheets to treat xeroderma pigmentosum, which was via decreasing the damage of UV light on the skin surface. Moreover, the relative problems of the theory and the experiments in manuscript have been addressed point by point. After revision this work is comprehensive, I recommend publication in the present form.